# Incident *Chlamydia trachomatis* Infection in a High School Population

**DOI:** 10.3390/biology11091363

**Published:** 2022-09-17

**Authors:** M. Jacques Nsuami, Wato Nsa, Catherine L. Cammarata, David H. Martin, Stephanie N. Taylor

**Affiliations:** 1Section of Infectious Diseases, Department of Medicine, School of Medicine, Louisiana State University Health Sciences Center, New Orleans, LA 70112, USA; 2Department of Medical Informatics, School of Community Medicine, University of Oklahoma Health Sciences Center, Tulsa, OK 74135, USA; 3Department of Epidemiology, School of Public Health & Tropical Medicine, Tulane University Health Sciences Center, New Orleans, LA 70112, USA

**Keywords:** adolescents, disease surveillance, epidemiology, parameter estimates, screening recommendations, sexually transmitted diseases

## Abstract

**Simple Summary:**

After a pathogen that causes disease has been introduced in a human population, an understanding of that disease in the population depends on knowing how many people in the population have the disease (prevalence), how fast the disease spreads from one person to another within the population (incidence), and how long the disease remains in an individual once acquired (duration). Infections caused by *Chlamydia trachomatis* are the most common sexually transmitted bacterial infections in the world. Individuals younger than 25 are the most affected due to the patterns of their sexual activity. In most individuals, these infections do not produce symptoms. Thus, affected individuals usually are not prompted to seek care, and most cases can only be detected through screening. Screenings for chlamydia in United States schools have given an indication of how many adolescents in the population might have chlamydia. In this study, we assessed how quickly chlamydia is acquired within the adolescent population. We determined that 14–19-year-old adolescents are acquiring chlamydia at a pace of 6.6 cases per 100 person-years for boys and 11.9 cases per 100 person-years for girls. Male and female students are acquiring chlamydia within 10 and within 7 months, respectively.

**Abstract:**

Prospective cohort studies of sexually transmitted infections (STIs) are logistically impractical owing to time and expenses. In schools, students are readily available for school-related follow-ups and monitoring. Capitalizing on the logistics that society already commits to ensure regular attendance of adolescents in school, a school-based STI screening in New Orleans made it possible to naturally observe the occurrence of chlamydia and to determine its incidence among 14–19-year-old adolescents. Among participants screened repeatedly, we calculated incidence rates, cumulative incidence, and incidence times. Male (n = 3820) and female (n = 3501) students were observed for 6251 and 5143 person-years, respectively, during which 415 boys and 610 girls acquired chlamydia. Incidence rates per 100 person-years were 6.6 cases for boys and 11.9 cases for girls. In multivariable analysis, the adjusted hazard ratio was 5.34 for boys and 3.68 for girls if the student tested positive for gonorrhea during follow-up, and 2.76 for boys and 1.59 for girls if at first participation the student tested positive for chlamydia, and it increased with age among boys but not among girls. In joinpoint trend analysis, the annual percentage change in the incidence rate was 6.6% for boys (95% CI: −1.2%, 15.1%) and 0.1% for girls (95% CI: −5.3%, 5.7%). Annual cumulative incidence was 5.5% among boys and 8.6% among girls. Median incidence time was 9.7 months for boys and 6.9 months for girls. Our findings can be used to refine assumptions in mathematical modeling and in cost analysis studies of *C. trachomatis* infection, and provide strong evidence in support of annual chlamydia screening for adolescent boys.

## 1. Introduction

In the study of the distribution of disease in specified populations, two measures are used to determine the occurrence of disease: prevalence, which counts the number of cases of existing disease in the population at a given time, and incidence, which determines the pace at which a new disease develops within the population [1,2,3]. Infection with *Chlamydia trachomatis* is the most common notifiable disease reported in the United States. In 2019, there were 1,808,703 cases reported to the Centers for Disease Control and Prevention (CDC), which, excluding cases with unknown gender, included 644,337 male and 1,160,470 female cases [4]. As in previous years, the rates of reported chlamydia per 100,000 people were highest among individuals younger than 25, with 1009 male cases and 3334 female cases aged 15 to 19, and 1871 male cases and 4109 female cases aged 20 to 24 [4].

Infections caused by *C. trachomatis* can result in urethritis in men and cervicitis in women, with repeat episodes, which are particularly common in adolescents and young adults, increasing the risks of serious complications in women, including pelvic inflammatory disease, chronic pelvic pain, ectopic pregnancy, susceptibility to co-infection with human papillomavirus, and tubal infertility [5,6]. The lifetime medical cost of infections with *C. trachomatis* that were acquired in 2018 in the United States through sexual contacts by individuals aged 15 to 24 was estimated to be approximately $0.5 billion, 91% of which incurred by women [7]. 

With the development in the early 1990s of nucleic acid amplification tests (NAATs) for the detection of sexually transmitted infections (STIs) in laboratory specimens that could be obtained without the need to perform clinical examination [8], programs of widespread screening for *C. trachomatis* and *Neisseria gonorrhoeae* were implemented in United States public high schools in New Orleans, New York City, and Philadelphia, in addition to smaller-scale screenings in Chicago, Detroit, Pittsburg, San Francisco, and Washington, DC [9]. A review of these programs commissioned by the CDC reported a prevalence of chlamydia among high school students that ranged between 2.5% and 7.0% in boys and between 8.1% and 13.1% in girls, when smaller-scale screening data from San Francisco were excluded [9]. It is worth noting that the upper limits of these ranges—7.0% for male students and 13.1% for female students—were from New Orleans [10]. These programs had not reported on the long-term incidence of chlamydia.

The purpose of the present study was to determine the incidence of chlamydia among students tested during 10 school years in the New Orleans STI screening program [10,11]; approximately half of the participants were boys. Insight into the pace at which students acquire chlamydia should contribute to better characterize the epidemiology of *C. trachomatis* infection in the general adolescent population [1,2,3,12,13] and to assess evidence for guiding chlamydia screening recommendations for adolescent boys [14].

## 2. Materials and Methods

### 2.1. Setting and Design

Between school years 1995–1996 and 2004–2005, students in a New Orleans public school district were offered chlamydia screening using NAATs with urine specimens [10,11]. All students in the participating schools were eligible for testing if they had consent, regardless of sexual activity or STI symptoms. Each year, parental consent was solicited for each student younger than 18. Students 18 years old or older gave their own consent in writing. The Institutional Review Board of the Louisiana State University Health Sciences Center in New Orleans approved and continually reviewed the screening protocol (LSUHSC-NO IRB#: 3675).

Two opportunities of screening were offered during each of the first two years and one opportunity afterward [10], for a total of 12 screening opportunities from January 11, 1996 to May 16, 2005. During specimen collection, students who had consent and were willing to be tested provided approximately 30 mL of a first-catch urine specimen. Testing was by a commercially available NAAT. In 1995–1996, specimens were initially tested by the Amplicor Chlamydia Test (Roche Molecular Systems, Branchburg, NJ, USA) then by the LCx Probe System (Abbott Laboratories, Abbott Park, IL, USA) until 1999–2000 [15]; from 2000–2001 to 2004–2005, the BDProbeTec ET System (Becton Dickinson, Sparks, MD, USA) was used [16]. From 1996–1997 to 2004–2005, each specimen collected was also tested for *N. gonorrhoeae* [10]. A positive test defined infection. Chlamydia infection was treated with 1 g oral azithromycin under direct observation. Treatment, counseling, and partner notification and referral in this screening program have been previously described in detail [10,11].

### 2.2. Study Subjects and Eligibility

During the 10-year period, a cumulative 35,974 students enrolled in at least 3 and as many as 13 participating schools [10,11]. Of these, 20,224 were tested at least once, and among them 7949 tested more than once. They were eligible for incidence calculations. We excluded 628 students who, at first participation, were younger than 14 (n = 605), older than 19 (n = 22), or whose date of birth was missing (n = 1). The remaining 7321 students (3820 boys and 3501 girls) aged 14 to 19 at first participation in screening and who were tested more than once constituted the subjects of this analysis. Study subjects were 98.9% African American (n = 7241) as were 93.8% (n = 33,746) of all students enrolled in the participating schools from 1995–1996 to 2004–2005.

### 2.3. Data Analysis

Among the students eligible for analysis, we calculated incidence rates, cumulative incidence, and incidence times [1,2,3]. 

#### 2.3.1. Incidence Rates Denominators

Between 1995–1996 and 2004–2005, students were available for testing during different and unequal times because each year a class completed and another class started high school. During the years they enrolled in participating schools, students could take part or not take part in one or the other screening opportunity, students could transfer to or from participating schools, and a school that participated for a year or years may not have participated at other times. To fully account for the unequal times each student contributed to the screening program, person-time denominators were used for calculating incidence rates [1,2,3]. For students whose first test was negative and did not contract chlamydia during follow-up, person-time at risk was the interval between their first and last tests. For students whose first test was positive and for whom the date of treatment was known, person-time at risk started the day of treatment. If the date of treatment was not available, the student was considered to have cleared the infection if the following test became negative or to have remained infected if the following test was again positive. If infection was cleared with no documentation of treatment, clearance was considered to have occurred halfway between the date the cleared infection was detected and the date of the following negative test. Among students who acquired chlamydia, because the time of infection could not be determined precisely, infection was estimated to have been acquired at the midpoint between the most recent negative test or the date of treatment and the following positive test [2]. Thus, person-time at risk for students who acquired chlamydia was the time from the date of the initial negative test or, if known, the date of treatment for a previous positive test, to the date of the last negative test before the positive test, plus half the time from the date of the most recent negative test to the date of the positive test. Person-time computations were performed using SPSS statistical software (IBM SPSS Statistics, Version 25) (SPSS Inc., Chicago, IL, USA).

#### 2.3.2. Cumulative Incidence Denominators

Cohorts of students tested during 2, 3, and 4 consecutive school years could be identified in the screening program database [11]. For these fixed and closed cohorts [1,3], the numbers of students at their initial participation were used as denominators for calculating cumulative incidence during the subsequent year(s) they were consecutively tested.

#### 2.3.3. Incidence Numerators

Because during follow-up only the first positive test was used to determine incident infection, observation for students who acquired chlamydia was censored at the time they were estimated to become infected. Thus, the numerators for incidence calculations were the numbers of students who acquired chlamydia, which were also identical to the numbers of incident infections in this report [2]. 

#### 2.3.4. Incidence Time Calculations

Among students who acquired chlamydia, incidence time was the number of days between the date of the initial negative test or the date of treatment for a previous positive test and the acquisition of the incident infection [2], i.e., the midpoint between their most recent negative test or the date of treatment and the positive test.

#### 2.3.5. Statistical Analyses

Analyses were performed separately by gender. For each gender, we computed binomial proportion 95% confidence intervals (CI) of incidence estimates using standard methods. Probability of incident chlamydia was estimated using the time-to-event analysis and summarized in a Kaplan–Meier curve (SAS, Version 9.4). We used multivariable Cox proportional hazard regression models to assess the independent contributions of risk factors on chlamydia incidence during follow-up [3] (SAS, Version 9.4). Incidence rate trends over the 11 follow-up screenings were analyzed using the National Cancer Institute Joinpoint regression software [17]. The statistical significance threshold was set at the 2-sided *p*-value of 0.05.

## 3. Results

### 3.1. Incidence Rates

The 7321 study subjects contributed 11,394 person-years of time at risk: 6251 person-years for boys and 5143 person-years for girls. During follow-up, 415 boys and 610 girls acquired chlamydia. The incidence rate was 6.6 cases/100 person-years for boys (415/6251; 95% CI: 6.0, 7.3) (Table 1), and 11.9 cases/100 person-years for girls (610/5143; 95% CI: 11.0, 12.8) (Table 2). The female-to-male incidence rate ratio (IRR) was 1.80.

Among male students, incidence rates were lower and similar for the 14- and 15-year age cohorts (3.4–4.8 cases/100 person-years) and higher and similar for the 16- through 19-year age cohorts (8.1–9.4 cases/100 person-years) (Table 1). There were no statistically significant differences in incidence rates between age cohorts among female students (Table 2).

Students whose chlamydia test results at first participation were positive had significantly higher incidence rates compared to those whose chlamydia test results at first participation were negative (boys: 22.5 (95% CI: 16.6, 29.5) vs. 6.2 (95% CI: 5.6, 6.8) cases per 100 person-years, IRR = 3.6, Table 1; girls: 24.4 (95% CI: 20.3, 28.9) vs. 10.8 (95% CI: 9.9, 11.7) cases per 100 person-years, IRR = 2.3, Table 2).

Among male students who had a positive gonorrhea test result during follow-up, the incidence rate was 36.9 cases/100 person-years (95% CI: 28.9, 45.4), and the incidence rate was 6.0 cases/100 person-years (95% CI: 5.4, 6.6) among male students who did not test positive for gonorrhea (IRR = 6.1) (Table 1). Among female students who had a positive gonorrhea test result during follow-up, the incidence rate was 40.9 cases/100 person-years (95% CI: 35.2, 46.7), and the incidence rate was 10.1 cases/100 person-years (95% CI: 9.3, 11.0) among female students who did not test positive for gonorrhea during follow-up (IRR = 4.0) (Table 2).

Male students’ incidence rates increased from 5.3 to 7.1 and 8.4 cases/100 person-years when testing was exclusively by the Abbott LCx assay, by both the Abbott and the BD ET assays, and exclusively by the BD ET assay, respectively (Table 1). Female students’ incidence rates were 11.2 cases/100 person-years when testing was exclusively by the Abbott LCx or by both assays, and 14.0 cases/100 person-years when testing was exclusively by the BD ET assay (Table 2).

In both male and female students’ multivariable analyses (Table 3), the risk of acquiring chlamydia was highest if the student had a positive gonorrhea test result during follow-up (adjusted hazard ratio (aHR): 5.34; 95% CI: 3.97, 7.17 for boys; aHR: 3.68; 95% CI: 2.98, 4.55 for girls). This was followed by whether at first participation the student tested positive for chlamydia (aHR: 2.76; 95% CI: 1.97, 3.88 for boys; aHR: 1.59; 95% CI: 1.27, 2.00 for girls). The risk of acquiring chlamydia increased significantly with age among male students (aHR: 1.31; 95% CI: 1.19, 1.44) but not among female students (aHR: 1.04; 95% CI: 0.96, 1.13).

In joinpoint trend analysis, the trends in incidence rates over time did not change significantly. The annual percentage change in the incidence rate was 6.6% (95% CI: −1.2%, 15.1%; *p* = 0.091) for boys and 0.1% (95% CI: −5.3%, 5.7%; *p* = 0.980) for girls (Figure 1 and Table 4).

### 3.2. Cumulative Incidence

Table 5 shows the cumulative incidence for fixed cohorts of students followed during consecutive years after their initial screening participation. There were 1571 (824 boys and 747 girls), 729 (402 boys and 327 girls), and 311 (170 boys and 141 girls) study subjects tested during 2, 3, and 4 consecutive years, contributing 1, 2, and 3 years of follow-up, respectively (average follow-up: 2 years). In these fixed cohorts, 153 of 1396 boys and 208 of 1215 girls acquired chlamydia. The annual cumulative incidence was 5.5% among boys ((153/1396 × 2); 95% CI: 4.7%, 6.4%) and 8.6% among girls ((208/1215 × 2); 95% CI: 7.5%, 9.7%). The female-to-male annual cumulative incidence ratio was 1.56.

### 3.3. Incidence Times

In male students, the median incidence time was 9.7 months (interquartile range (IQR): 12.8 months), decreasing with age; 6 months in cohorts 17 years old and older; and 6.6 months if at first participation the student tested positive for chlamydia (Table 1). In female students, the median incidence time was 6.9 months (IQR: 9.2 months), decreased with age; ranged between 5.5 and 9 months in cohorts 15 years old and older; and was 6.1 months if at first participation the student tested positive for chlamydia (Table 2). In both genders, the probability of incident chlamydia increased with longer follow-up time (Figure 2).

## 4. Discussion

To determine the incidence of a disease, it is necessary to follow a cohort of individuals initially free from the disease and observe the occurrence of the disease within the cohort. Cohort studies, particularly in their prospective designs, typically involve following large numbers of individuals for many years to observe the occurrence of the disease, which make them time-consuming and expensive [1,2,3]. The time and expenses for following large numbers of individuals over time have made prospective cohort studies of STIs logistically impractical. In schools, students are readily available for school-related follow-ups and monitoring. A school-based STI screening, capitalizing on the logistics that society already commits to ensure regular attendance of adolescents in school, made it possible to naturally observe the occurrence of chlamydia and to determine its incidence among students who participated repeatedly [10,11]. 

The individual longitudinal data gathered from participants in the New Orleans school-based STI screening show a high incidence of chlamydia, especially considering that participants were healthy adolescents who were not seeking care at the time of testing. The risk of acquiring chlamydia was significantly higher among girls than among boys, and increased with age among boys and with follow-up time in both genders. Sexual exposure (gonorrhea infection detected during follow-up and a positive chlamydia test result at first participation) was a stronger risk factor in predicting incident chlamydia than demographic factors. Incidence estimates in this study may be conservative because all students were tested for STIs regardless of sexual activity, which implies that some participants may not have been sexually exposed between screening [18].

Testing for *C. trachomatis* in this screening program was by a commercially available NAAT. The first 444 specimens were tested by the Roche Amplicor chlamydia assay [10], then by the Abbott LCx assay, until reports of performance issues that later affected its availability on the market began to emerge [19,20,21]. As local laboratories moved from the Abbott LCx assay to the BD ET assay, chlamydia incidence among boys whose urine specimens were tested by both assays were between the rates observed when testing was by the Abbott LCx and when testing was by the BD ET system, but not among girls (Table 1, Table 2 and Table 3). Switching from the LCx assay in 1999–2000 to the BD ET assay in 2000–2001 was associated with increased chlamydia positivity [11], which, however, was demonstrated not to indicate increased prevalence [10]. In the current study, the 95% confidence interval of the annual percentage change of the upward slope observed in the incidence rate trend among boys also included 0 (Table 4). This indicates that over time, the switch from the LCx assay to the BD ET assay did not significantly affect the trend in incidence rates among boys. Overall, our findings indicate that chlamydia incidence in the high school male and female student population remained stable between 1996 and 2005.

### 4.1. Strengths

The observation of 3820 male students developing incident chlamydia over 6253 person-years added data to our understanding of the epidemiology of genital infection with *C. trachomatis* in the general adolescent population. Findings from this study also contributed evidence that can be assessed for guiding recommendations for screening male adolescents for chlamydia.

The minimal female-to-male ratios of chlamydia incidence in this study are 1.51 for incidence rates and 1.17 for annual cumulative incidence, when the ratios are determined by dividing the girls’ lower limits by the boys’ upper limits of their 95% confidence intervals. These findings have direct implications on how estimates of population distributions of chlamydial infections are calculated. Because data for determining chlamydia incidence are usually unavailable, it became customary to estimate population incidence using available prevalence data and by assuming that infections are of known duration in a steady-state population [22,23,24]. Under steady-state assumptions, prevalence of a disease is directly proportional to its incidence and its duration, and that if two of these three measures are known, the third can be calculated [1,2,13]. With genital chlamydial infections, however, no two of these three measures are reliably available. The duration of infection is generally unknown, and the robustness of prevalence data across populations is often uncertain. Thus, the reported incidence of chlamydia in populations is entirely based on estimates that are based on assumptions, not data. One consequence of this lack of reliable measures is that the female-to-male ratio of chlamydia incidence has been assumed to be equal to one, suggesting equal incidence between males and females [22,24]. That girls had an incidence rate 80% higher and an annual cumulative incidence 56% higher than boys in this study represents a significant contribution to our understanding of the epidemiology of genital infection with *C. trachomatis* in the adolescent population [12,13]. Incidence measures for genital infection with *C. trachomatis* among adolescent boys and girls are not equal. These findings provide parameter values for mathematical modeling studies that estimate population distributions of genital *C. trachomatis* infections [25,26] or their costs to society [7,27].

This longitudinal study also created the opportunity of estimating time to the acquisition of chlamydia. Male and female students aged 17 to 19 acquired chlamydia at a median time of 6 months, with incidence time declining sharply from ages 14 to 17 in both genders. In 2005, 22%, 72%, and 128% more 10th, 11th, and 12th graders in the United States, respectively, reported four or more lifetime sex partners compared to 9th graders [18]. The sharp decline in chlamydia incidence time from ages 14 to 17 in this study therefore reflects a steady increase in sexual exposure among high school students within this age range [18]. A consequence of this increase in sexual exposure was the documentation that male and female students who tested positive for chlamydia at first participation became reinfected after median intervals of 6.6 and 6.1 months, respectively (Table 1 and Table 2).

### 4.2. Limitations

By estimating the time of infection at the midpoint between the most recent negative test and the following positive test [2], person-time denominators might have been systematically increased if all or most infections were acquired near the collection date of the positive specimen, or decreased if all or most infections were acquired near the collection date of the negative specimen. Although these would not have affected incidence trends and cumulative incidence calculations, both scenarios would have underestimated or overestimated incidence rate calculations, respectively. However, it is reasonable to assume a random distribution of the time of infection within the interval between screenings rather than a clustering of acquisition of almost all infections at one end of the interval or the other. The midpoint is therefore the best single point estimate of that timing and it provided an unbiased estimate of person-time at risk for students who acquired chlamydia during follow-up [2]. 

This study assumed a complete tally of infections acquired during follow-up. However, it remains possible that chlamydia could have been diagnosed outside of our program among study subjects that were not counted as incident infections in this analysis. Such incidents were nonetheless unlikely to have occurred in a well-defined group of students to affect trends over time. Additionally, such incidents could only add to an already high incidence in this population.

Because consent was required each year, students could not participate during the year(s) for which they did not have consent even if they wanted to, possibly accounting for testing gaps among some study subjects [11]. Longer observation time was associated with increased probability of acquiring chlamydia (Figure 2).

## 5. Conclusions

A sufficiently large number of high school adolescents were observed developing chlamydia during a 10-year period in New Orleans, which contributed data that can be used to refine assumptions in mathematical modeling and in cost analysis studies of genital infection with *C. trachomatis*. Incidence of chlamydia among adolescent girls is not equal to incidence among adolescent boys [12,22,24]. These data additionally contribute evidence that can be assessed to update recommendations for chlamydia screening [14]. While incidence times from this study support current recommendations for annually screening adolescent girls for chlamydia [28,29,30,31], they also provide strong evidence in support of annual chlamydia screening for adolescent boys.

## Figures and Tables

**Figure 1 biology-11-01363-f001:**
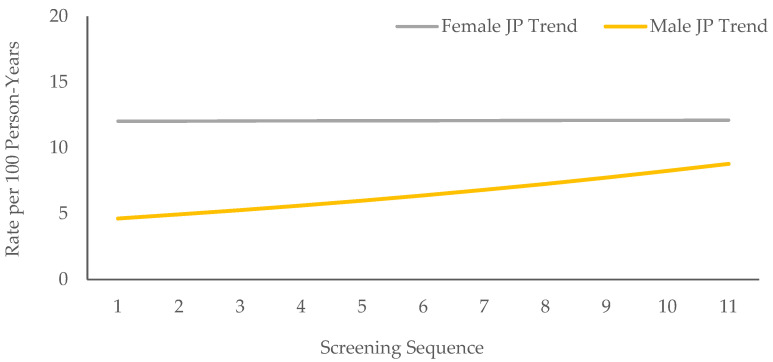
Joinpoint (JP) trend analyses of incidence rates over the 11 follow-up screenings.

**Figure 2 biology-11-01363-f002:**
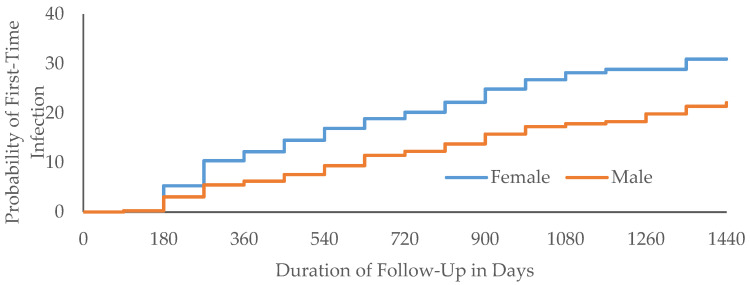
Time (days) to incident chlamydia.

**Table 1 biology-11-01363-t001:** Number of male students observed (n = 3820), person-times at risk, number of students with incident chlamydia, incidence rates, and incidence time among students who acquired incident infection (n = 415).

	Number Observed	Person-Years at Risk	Students with Incident Chlamydia (n)	Incident Cases per 100 Person-Years (95% CI)	Incidence Time (Months) Median (IQR)
Total	3820	6251	415	6.6 (6.0, 7.3)	9.7 (12.8)
Age at first participation (years)					
14	299	703	24	3.4 (2.2, 5.0)	25.9 (12.2)
15	1122	2307	111	4.8 (4.0, 5.8)	17.4 (17.1)
16	1179	1881	152	8.1 (6.9, 9.4)	11.3 (11.4)
17	808	954	90	9.4 (7.7, 11.5)	5.9 (3.5)
18 or 19	412	405	38	9.4 (6.7, 12.7)	6.0 (3.4)
Chlamydia test result at first participation					
Positive	173	173	39	22.5 (16.6, 29.5)	6.6 (10.2)
Negative	3647	6078	376	6.2 (5.6, 6.8)	10.5 (12.9)
Had a positive gonorrhea test during follow-up					
Yes	104	141	52	36.9 (28.9, 45.4)	9.6 (12.8)
No	3713	6105	363	6.0 (5.4, 6.6)	9.7 (12.8)
Nucleic acid amplification test					
PCR/LCx assay (1995–1996 to 1999–2000)	1805	2543	134	5.3 (4.4, 6.2)	6.5 (11.2)
PCR/LCx and BD assays (prior to and after 2000)	1007	2290	162	7.1 (6.1, 8.2)	14.7 (17.2)
BD assay (2000–2001 to 2004–2005)	1008	1418	119	8.4 (7.0, 10.0)	7.6 (11.1)

CI, confidence interval; IQR, interquartile range.

**Table 2 biology-11-01363-t002:** Number of female students observed (n = 3501), person-times at risk, number of students with incident chlamydia, incidence rates, and incidence time among students who acquired incident infection (n = 610).

	Number Observed	Person-Years at Risk	Students with Incident Chlamydia (n)	Incident Cases per 100 Person-Years (95% CI)	Incidence Time (Months) Median (IQR)
Total	3501	5143	610	11.9 (11.0, 12.8)	6.9 (9.2)
Age at first participation					
14	280	537	55	10.2 (7.8, 13.1)	12.7 (19.4)
15	1189	2093	227	10.9 (9.6, 12.3)	8.7 (11.3)
16	1186	1623	211	13.0 (11.4, 14.7)	6.8 (8.4)
17	596	648	85	13.1 (10.6, 16.0)	6.2 (3.1)
18 or 19	250	242	32	13.2 (9.2, 18.2)	5.5 (2.7)
Chlamydia test result at first participation					
Positive	354	405	99	24.4 (20.3, 28.9)	6.1 (6.5)
Negative	3147	4738	511	10.8 (9.9, 11.7)	7.5 (9.8)
Had a positive gonorrhea test during follow-up					
Yes	240	296	121	40.9 (35.2, 46.7)	6.4 (10.8)
No	3253	4839	489	10.1 (9.3, 11.0)	7.3 (8.9)
Nucleic acid amplification test					
PCR/LCx assay (1995–1996 to 1999–2000)	1737	2216	248	11.2 (9.9, 12.6)	6.0 (6.2)
PCR/LCx and BD assays (prior to and after 2000)	833	1709	192	11.2 (9.8, 12.8)	12.6 (13.6)
BD assay (2000–2001 to 2004–2005)	931	1217	170	14.0 (12.1, 16.0)	6.5 (6.0)

CI, confidence interval; IQR, interquartile range.

**Table 3 biology-11-01363-t003:** Cox proportional regression analyses of incidence rates among male and female students.

	Male Students		Female Students	
Potential Risk Factor	Adjusted Hazard Ratio (95% CI)	*p*-Value	Adjusted Hazard Ratio (95% CI)	*p*-Value
Age at first participation	1.31 (1.19–1.44)	<0.001	1.04 (0.96–1.13)	0.317
Positive chlamydia at first participation	2.76 (1.97–3.88)	<0.001	1.59 (1.27–2.00)	<0.001
Positive gonorrhea during follow-up	5.34 (3.97–7.17)	<0.001	3.68 (2.98–4.55)	<0.001
PCR/LCx and BD assays vs. PCR/LCx assay	1.36 (1.08–1.73)	0.010	1.00 (0.82–1.21)	0.963
BD assay vs. PCR/LCx assay	1.37 (1.07–1.76)	0.013	1.18 (0.97–1.44)	0.094

BD assay (2000–2001 to 2004–2005); CI, confidence interval; PCR/LCx assay (1995–1996 to 1999–2000); PCR/LCx and BD assays (prior to and after 2000).

**Table 4 biology-11-01363-t004:** Annual percentage change (APC) in incidence rates over time.

	APC % (95% CI)	*p*-Value
Male Students	6.6 (−1.2, 15.1)	0.091
Female Students	0.1 (−5.3, 5.7)	0.980

CI, confidence interval.

**Table 5 biology-11-01363-t005:** Cumulative incidence among 1396 male and 1215 female students tested for chlamydia during 2, 3, and 4 consecutive school years (total = 2611).

Years Tested Consecutively	Years Followed up from Year 1	Number Observed (n)	Number with Incident Infection (n)	Cumulative Incidence ^a^ (%)	Annual Incidence ^b^ % (95% CI)
**Male Students**	**2**	**1396**	**153**	**11.0**	**5.5 (4.7, 6.4)**
2	1	824	61	7.4	7.4 (5.7, 9.4)
3	2	402	60	14.9	7.5 (5.7, 9.5)
4	3	170	32	18.8	6.3 (4.3, 8.7)
**Female Students**	**2**	**1215**	**208**	**17.1**	**8.6 (7.5, 9.7)**
2	1	747	102	13.7	13.7 (11.3, 16.3)
3	2	327	65	19.9	9.9 (7.8, 12.5)
4	3	141	41	29.1	9.7 (7.0, 12.9)

CI, confidence interval. ^a^ Cumulative incidence = (incident infections/number observed) × 100. ^b^ Annual incidence = cumulative incidence/years followed up from year 1.

## Data Availability

The data presented in this study are available upon request from the corresponding author. The data are not publicly available due to data source sensitivities and confidentiality of patient health information.

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
