# Peer review of "Incident Chlamydia trachomatis Infection in a High School Population"

_biology, 2022, doi:10.3390/biology11091363_

Round 1

Reviewer 1 Report

The Authors’ manuscript is an interesting large prospective cohort study with the aim of determining the incidence of C. trachomatis infection among high school students during 10 school years in New Orleans Public School district. Incidence rates, cumulative incidence and incidence times were calculated.

The study is well written, Materials and Methods are scientifically sound and the statistical analysis is appropriate. Strengths and limitations are clearly highlighted in the Discussion section.

I have, however, some minor criticisms:

-          In the Introduction section, it would be useful to the reader to add a section describing the clinical importance of C. trachomatis infection, such as its manifestations, chronic complication, and impact on public health, allowing the reader to better frame the extent of the problem that the Authors aimed to address in their study;

-          In Materials and Methods section, paragraph 2.1, it should be specified the number of schools included in the study and from which school students were enrolled;

-          In Results and Discussion sections, there is no mention whether students diagnosed for a C. trachomatis infection showed symptoms, or whether they were, instead, asymptomatic. Given the impact of asymptomatic infections in underestimating the real prevalence and incidence of C. trachomatis (Di Pietro M et al., 2019 doi: 10.3390/microorganisms7050140; O’Connell and Ferone, 2017 doi: 10.15698/mic2016.09.525), it would be interesting to integrate the Authors’ analysis with data on chlamydial clinical manifestations, if available.

Reviewer 2 Report

Nsuami et al have highlighted the incident Chlamydia trachomatis Infection in a High School Population in New Orleans. USA.

The draft has a few interesting points for the readers but lacks information on important information such as the characteristics of the students involved (clinical presentations, no of sexual partners, risky behaviour etc2), and predictors. My major concern is on the data. It has been more than a decade!

Several articles have already been published which are more informative and several recommendations have been proposed including the annual screening of STI for male and female students. Thus, the gap is inapparent.

List of papers on similar topics:

Braverman PK. Incident Chlamydia trachomatis infections among inner-city adolescent females. Clin Pediatr (Phila). 2000 Jun;39(6):378. doi: 10.1177/000992280003900612. PMID: 10879944

Burstein GR, Gaydos CA, Diener-West M, Howell MR, Zenilman JM, Quinn TC. Incident Chlamydia trachomatis Infections Among Inner-city Adolescent Females. JAMA. 1998;280(6):521–526. doi:10.1001/jama.280.6.521

Mahboobeh Safaeian, Koen Quint, Mark Schiffman, Ana Cecilia Rodriguez, Sholom Wacholder, Rolando Herrero, Allan Hildesheim, Raphael P. Viscidi, Wim Quint, Robert D. Burk, Chlamydia trachomatis and Risk of Prevalent and Incident Cervical Premalignancy in a Population-Based Cohort, JNCI: Journal of the National Cancer Institute, Volume 102, Issue 23, 1 December 2010, Pages 1794–1804, https://doi.org/10.1093/jnci/djq436

Detection of gonorrhoea is also lacking. Enrolment of the students is unclear with no exclusion and inclusion criteria.

What is the current policy on STI screening in the schools? How can their findings affect the policy?

Reviewer 3 Report

Dear Authors your manuscript titled "Incident Chlamydia trachomatis Infection in a High School Population" is very interesting due to the impact of sexually transmitted diseases on the young population and the need to provide detailed information at school

I suggest you provide in the introduction of the paper some more information on the disease related to C. trachomatis infection, just to better frame the impact on the health of adolescents and young people. 
